# Neuroprotective Effects of *Euonymus alatus* Extract on Scopolamine-Induced Memory Deficits in Mice

**DOI:** 10.3390/antiox9050449

**Published:** 2020-05-22

**Authors:** Yunju Woo, Ji Sun Lim, Jisun Oh, Jeong Soon Lee, Jong-Sang Kim

**Affiliations:** 1School of Food Science and Biotechnology (BK21 plus), Kyungpook National University, Daegu 41566, Korea; dndbswn123@naver.com (Y.W.); lzsunny@daum.net (J.S.L.); j.oh@knu.ac.kr (J.O.); 2Institute of Agricultural Science and Technology, Kyungpook National University, Daegu 41566, Korea; 3Forest Resources Development Institute of Gyeongsangbuk-do, Andong 36605, Korea; ljs7942@korea.kr

**Keywords:** *Euonymus alatus*, antioxidant, BDNF, cognition improvement, neuroprotective, HT22 cells, animal model

## Abstract

*Euonymus alatus* is considered to elicit various beneficial effects against cancer, hyperglycemia, menstrual discomfort, diabetic complications, and detoxification. The young leaves of this plant are exploited as food and also utilized for traditional medicine in East Asian countries, including Korea and China. Our preliminary study demonstrated that ethanolic extract from the *Euonymus alatus* leaf (EAE) exhibited the strongest antioxidant enzyme-inducing activity among more than 100 kinds of edible tree leaf extracts. This study investigated whether EAE could attenuate the cognitive deficits caused by oxidative stress in mice. Oral intubation of EAE at 100 mg/kg bw or higher resulted in significant improvements to the memory and behavioral impairment induced via i.p. injection of scopolamine. Furthermore, EAE enhanced the expression levels of hippocampal neurotrophic factors such as brain-derived neurotrophic factor (BDNF) and glial cell-derived neurotrophic factor in mice, activated the Nrf2, and the downstream heme oxygenase-1 (HO-1) a quintessential antioxidant enzyme. As rutin (quercetin-3-*O*-rutinose) was abundantly present in EAE and free quercetin was able to induce defensive antioxidant enzymes in an Nrf2-dependent manner, our findings suggested that quercetin derived from rutin via the intestinal microflora played a significant role in the protection of the mouse hippocampus from scopolamine-induced damage through BDNF-mediated Nrf2 activation, thereby dampening cognitive decline.

## 1. Introduction

Nuclear factor erythroid-derived 2-related factor 2 (NRF2) is one of the transcription factors that stimulate the expression of a variety of detoxifying and/or antioxidant enzymes via interaction with the antioxidant response elements (AREs) [1,2]. Recent studies demonstrated that the activation of the Nrf2 signaling pathway could ameliorate cognitive defects in Alzheimer’s disease model mice [2,3,4]. The main features of Alzheimer’s disease (AD) include neuronal loss and synaptic dysfunction, which is attributable to oxidative stress mainly produced by amyloid-β(Aβ)-peptide. Although numerous compounds reportedly activate the Nrf2 signaling pathway, a few studies examined the potential of using a plant extract or other plant constituents to improve the cognitive function of mice in an AD animal model. Thus, we hypothesized that plant extract might be able to strongly activate the Nrf2 signaling pathway, which could attenuate memory impairment caused by scopolamine, a muscarinic cholinergic receptor antagonist that alters brain oxidative stress status [5]. Screening of more than 100 types of plant extracts for efficacy in the stimulation of the Nrf2 signaling pathway using mouse hippocampal HT22 cells transfected with ARE showed that the extract of edible *Euonymus alatus* leaves strongly induced antioxidant enzymes through the Nrf2 signaling pathway.

*Euonymus alatus* is a flowering plant that has been used as a medicinal plant by some Asian countries for centuries in the treatment of various conditions including cancer, hyperglycemia, menstrual discomfort, diabetic complications, and detoxification [6]. The leaves of this plant are registered as edible in the Korean Food Code. Some constituents, including rutin, kaempferol, and quercetin extracted from *Euonymus spp*. were reported to exhibit antioxidant activity [7,8].

This study investigated whether an extract created from *Euonymus alatus* leaves has the potential to improve ROS-induced cognitive dysfunction (which mimics AD) by activating the Nrf2 signaling pathway. 

## 2. Materials and Methods 

### 2.1. Preparation of the Euonymus alatus Leaf Extract (EAE)

*Euonymus alatus* leaves were supplied by the Forest Resources Development Institute of Gyeongsangbuk-do (Andong, Korea). *Euonymus alatus* leaves (100 g) were extracted with 2 L of 80% (*v/v*) ethanolic solution combined with shaking (150 rpm) at 25 °C for 24 h. The extract was filtered through filter paper (185 mm, Hyundai Micro, Seoul, Korea) and dehydrated by vacuum-evaporation (EYELA N-1000, Tokyo, Japan), followed by freeze-drying. The dried sample was stored at −20 °C until use.

### 2.2. Chemicals and Reagents

Dulbecco’s modified Eagle’s medium (DMEM), trypsin with ethylenediaminetetraacetic acid (trypsin-EDTA), and penicillin-streptomycin (10,000 U/mL/10,000 μg/mL; HyClone Laboratories Inc., Logan, TU, USA) were supplied from Welgene (Gyeongsan, Korea) and fetal bovine serum (FBS; Gibco^™^) was purchased from Thermo Fisher Scientific (Waltham, MA, USA). Reagents used in the gel electrophoresis and western blot were purchased from Bio-Rad (Hercules, CA, USA). Antibodies were purchased from Abcam (Cambridge, UK) and Santa Cruz Biotechnology (Dallas, TX, USA). l-glutamic acid monosodium salt monohydrate, 2,2′-azino-bis (3-ethylbenzothiazoline-6-sulphonic acid) (ABTS), 2,2-diphenyl-1-picrylhydrazyl (DPPH), and scopolamine hydrobromide were obtained from Sigma-Aldrich (St. Louis, MO, USA). D-Plus™ CCK cell viability assay kit was obtained from Dongin LS (Seoul, Korea).

### 2.3. Euonymus alatus Leaf DPPH and ABTS Radical Scavenging Assays

The radical scavenging activity of DPPH by the sample was determined according to a protocol described previously [9]. Briefly, an aliquot of the ethanolic extract (50 μL) from *Euonymus alatus* leaf was mixed with 200 μL of 0.2 mM methanolic DPPH radical solution and incubated at 37 °C for 30 min. The absorbance (Abs) was measured at 515 nm using a microplate reader (Sunrise; Tecan, Grödig, Austria). 

The ABTS radicals were generated by mixing 7 mM ABTS with 2.45 mM potassium persulfate in deionized water and subsequently incubating this solution in the dark at room temperature overnight. The Abs of the radical solution was adjusted to 0.70 ± 0.02 at a wavelength of 734 nm by diluting it with absolute ethanol. ABTS solution (90 μL) was dispensed in a 96-well microplate, and various concentrations of the sample extract (10 μL) were allowed to react with the radical solution for 5 min. The Abs of the reactant was observed at 734 nm using a microplate reader (Infinite 200; Tecan, Gröding, Austria) [10,11]. 

### 2.4. Determination of Rutin and Quercetin

As rutin and quercetin were reported to be the major constituents of the *Euonymus alatus* leaf, the quantities of those compounds in the EAE were determined by HPLC analysis. HPLC analysis was performed on the reversed-phase Gemini C18 column supplied by Phenomenex (250 × 4.6 mm, 5 μm, Torrance, CA, USA) with 1% acetic acid (40:15:45) MeOH:ACN:DW as the mobile phase and a flow rate of 1 mL/min. The column temperature was maintained at 30 °C throughout the analysis. The chromatograms were procured at a wavelength of 360 nm, and the injection volume was 10 μL. EAE was injected three times each, and the averages of the peak areas on the chromatograms were obtained and extrapolated to the standard curve of known concentrations of rutin and quercetin for quantification.

### 2.5. Measurement of Inhibitory Activity against Acetylcholine Esterase

The mouse cortical tissue was homogenized in PBS, centrifuged, and the supernatant was used as a source of acetylcholinesterase (AChE). The inhibitory activity of EAE against AChE was determined according to the procedures described previously [12,13].

### 2.6. Cell Culture

A mouse hippocampal neuronal HT22 cell line (generously provided by Professor Dong Seok Lee, Kyungpook National University, Daegu, Korea) was grown as a monolayer in DMEM (Welgene, Gyeongsan, Korea) supplemented with 10% (*v/v*) heat-inactivated FBS (Gibco^™^/Thermo Fisher Scientific, Waltham, MA, USA) and 1% penicillin-streptomycin solution (HyClone, Logan, UT, USA) and subcultured when the confluency reached approximately 70% as previously reported [14].

### 2.7. Cell Viability Test and Transcriptional Activity Assay

Neuronal HT22 and its antioxidant response element (ARE)-transfectant cells (2 × 10^3^ cells/well) were plated onto the 96-well culture plate in 10% (*v/v*) FBS-containing DMEM. After incubating for 24 h, HT22 cells were treated with EAE for another 24 h in the absence and presence of 5 mM glutamate. The cell viability was then assessed by the CCK cell viability assay kit (D-Plus™, Dongin LS, Seoul, Korea) as per the supplier’s protocol. The relative cell survival was expressed as the percentage of untreated cells. The Nrf2 transcriptional activation assay in which the HT22-ARE cell line, a transfectant carrying ARE and luciferase reporter gene was used, was performed according to the protocol previously reported [14].

### 2.8. Animal Experiment

C57BL/6J male mice (5 weeks old; 17–19 g) were supplied by Daehan Biolink (Eumseong, Korea). Mice were placed in polycarbonate cages (278 mm × 420 mm × 200 mm; ≤ 8 mice/cage) and allowed to adapt for 1 week and then kept under the following conditions; temperature of 25 °C; humidity 50% ± 5%; a 12 h shift of the light/dark cycles, light period starting at 9:00 a.m., with water and feed *ad libitum*. Mice were given ad libitum access to Chow food pellets (Daehan Biolink, Eumseong, Korea) and tap water. All animal studies were performed according to the guidelines of the Committee on the Care and Use of Laboratory Animals of the Kyungpook National University (Approval Number 2019-0096). Mice were randomly assigned to six groups with eight mice per group as shown in Table 1; (1) group treated with saline only; (2) group treated with saline + scopolamine (1 mg/kg BW); (3) group treated with donepezil (5 mg/kg BW) + scopolamine (1 mg/kg BW); (4) EAE (50 mg/kg BW) + scopolamine; (5) EAE (100 mg/kg BW) + scopolamine; (6) EAE (150 mg/kg BW) + scopolamine. Donepezil, an acetylcholinesterase inhibitor, and EAE were orally administered using a zonde daily for 2 weeks, whereas the scopolamine (1 mg/kg BW) was injected i.p. into mice. EAE was dissolved in 5% ethanol and 5% Tween-80 in 0.9% saline prior to oral administration. Memory deficits were generated in mice via i.p. injection of scopolamine 30 min before the behavioral tests. After finishing behavioral testing, the mice were sacrificed by asphyxiation in a CO_2_ chamber. The following were collected from the mice, the brain, liver, plasma, and spleen, and the organs or tissues were quickly frozen in liquid N_2_ and stored at −80 °C until further analyses. 

### 2.9. Animal Behavioral Testing

Behavioral tests including passive avoidance, Y-maze, Morris water tests were conducted to assess the effect of EAE on the association memory, working memory, spatial learning, and long-term memory, respectively, according to the procedures described previously [14,15,16].

### 2.10. Measurement of the Serum 8-Hydroxy-2′-Deoxyguanosine (8-OHdG) Level

8-hydroxy-2′-deoxyguanosine (8-OHdG) is the most commonly used biomarker for examining oxidative DNA damage. Whole blood was incubated for 30 min at room temperature, centrifuged at 4800 × *g* for 15 min using a 1730 MR centrifuge (Gyrogen, Gimpo, Korea), and then quantified via DNA damage ELISA kit (Enzo Life Sciences International Inc., Plymouth Meeting, PA, USA) according to the supplier’s protocol.

### 2.11. Determination of Lipid Peroxidation in Liver Tissues

Lipid peroxidation was evaluated as the level of malondialdehyde (MDA) in the homogenized liver tissue. The reaction of MDA with thiobarbituric acid (TBA) produces a thiobarbituric acid-reactive substance (TBARS) that could be measured with a colorimeter [17]. Liver tissues collected from the experimental groups were homogenized, centrifuged at 15,000× *g*, for 30 min, and the supernatant was subjected to the TBARS assay using an OxiSelect TBARS Assay Kit (Enzo Life Science, Inc., Farmingdale, NY, USA), as stated in the protocol supplied by the manufacturer.

### 2.12. Western Blotting

Dissected brain tissues were homogenized in the precooled lysis buffer (20 mM Tris–HCl, 145 mM NaCl, 10% glycerol, 5 mM EDTA, 1% Triton-X, and 0.5% Nonidet-P 40), with the protease inhibitor (Roche Diagnostics, Mannheim, Germany). The homogenized brain tissue was fractionated with cytoplasmic and nuclear proteins using a NE-PER^®^ nuclear and cytoplasmic extraction reagent (Thermo Fisher Scientific) following the supplier’s protocols. The protein contents in the extracted fractions were quantified by a Bradford assay, and equal amounts of the proteins were denatured by heating at 95 °C in sample loading buffer (62.5 mM Tris–HCl, pH 6.8, 2% SDS, 0.1% bromophenol blue, 5% β-mercaptoethanol, 20% glycerol), and separated on a 10% polyacrylamide gel. Then, the electrophoresis of proteins in the SDS-PAGE gel, a transfer of protein bands onto the PVDF membrane, antibody binding, visualization, and densitometry of protein bands on a gel were performed as described previously [14,18]. The primary antibodies used in this study were rabbit anti-Nrf2 (Abcam), rabbit anti-HO-1 (Abcam), mouse anti-PSD-95 (Santa Cruz Biotechnology), mouse anti-β-actin (Santa Cruz Biotechnology), and goat anti-lamin B (Santa Cruz Biotechnology).

### 2.13. Quantitative PCR Analysis

The mRNA expression of genes encoding brain-derived neurotrophic factor (*BDNF*), glial cell-derived neurotrophic factor (*GDNF*), and *N*-methyl-d-aspartate (*NMDA1*) in the hippocampus were determined by real-time PCR. Total RNA was isolated from the hippocampus using a column-based RNeasy Mini Kit (QIAGEN, Hilden, Germany) and was reverse-transcribed into cDNA using reverse transcriptase (M-MLV, Thermo Fisher Scientific) with oligo(dT)12–18 primer. The quantitative real-time PCR using SYBR Green was conducted to determine the relative expression levels for the respective genes, using LightCycler^®^ Multiplex Masters (Roche, Basel, Switzerland) with the appropriate primer sets (Table 2) on a LightCycler^®^ Nano Instrument (Roche). The transcript expression levels were normalized to the expression level of the β-actin [19].

### 2.14. Histology

The brain tissues from the sacrificed mice were quickly rinsed in PBS and fixed in a 3.7% (*v/v*) formalin solution. Each tissue was placed in an embedding cassette (Simport, Beloeil, QC, Canada). Following rinsing in PBS for at least 30 min, the brain tissues were soaked sequentially in 60%, 70%, 80%, 90%, 95%, and 100% ethanol for 1 h each for dehydration, and subsequently in xylene for 2 h. Then, the xylene, an infiltration agent, was replaced with paraffin. The brain tissue in each cassette was then dipped in a melted paraffin solution at 65 °C for 1 h, which was repeated in triplicate. The paraffin blocks in which the brain tissue was embedded were sectioned to a 5 μm thickness using a microtome (RM-2125 RT; Leica, Nussloch, Germany). The sections were placed on Superfrost Plus microscope slides (Marienfeld, Lauda-Königshofen, Germany), air-dried at 37 °C for 12 h, and then stained with hematoxylin and eosin dyes. Following mounting using a mounting solution (Dako Fluorescent Mounting Medium; Dako, Glostrup, Denmark), the CA1 region of the hippocampus was observed under an optical microscope (Eclipse 80i, Nikon, Tokyo, Japan).

### 2.15. Statistical Analysis

The obtained data were expressed as the mean ± SD and analyzed by one-way analysis of variance (ANOVA), followed by Duncan’s multiple range test, using SPSS statistics 22 software (SPSS Inc., Chicago, IL, USA). *p*-values less than 0.05 were considered statistically significant. Statistically significant differences were marked using different alphabetical letters on values.

## 3. Results

### 3.1. Radical Scavenging Activity of EAE

EAE exhibited relatively strong DPPH and ABTS radical scavenging activities in a dose-dependent manner, as shown in Figure 1. Specifically, the DPPH and ABTS radical scavenging activities of EAE at 200 μg/mL were higher than 50%.

### 3.2. The Cytoprotective Effects of EAE

When the HT22 cells were exposed to 5 mM glutamate, cell viability was significantly reduced, as shown in Figure 2. In contrast, EAE in the range of 0.78–6.25 μg/mL suppressed the glutamate-induced cell growth inhibition and increased cell viability in a dose-dependent fashion in the presence of glutamate. The viability of the mouse hippocampal HT22 cells treated with EAE at the dose of 3.125 μg/mL or higher in combination with 5 mM glutamate was found to be higher than the negative control, which was not treated with 5 mM glutamate, demonstrating that the EAE had protected the neural cells from glutamate-induced cytotoxicity.

### 3.3. Effect of EAE on Scopolamine-induced Cognitive Impairment in Mice

Mice were treated with scopolamine (1 mg/kg bw) in combination with various doses of EAE, followed by the Y-maze test, passive avoidance test, and the Morris water maze test to examine whether EAE could restore the scopolamine-induced cognitive deficits of mice. The fear-motivated tests including the passive avoidance test were commonly used to investigate the short-term or long-term memory of small laboratory animals like rats and mice. These tests showed that the scopolamine-induced memory deficit of the mice was partially restored by the oral administration of EAE at a dose of 50 mg/kg bw and completely at a dose of 100 mg/kg bw or higher (Figure 3A).

The Morris water maze test was also conducted to investigate whether EAE produced a beneficial effect on the spatial learning and memory defects induced by the scopolamine injection. The injection with scopolamine alone had significantly impaired the cognitive function of mice. However, oral administration of EAE at a dose of 150 mg/kg bw prior to the scopolamine injection completely restored the spatial learning and memory function of mice impaired by the scopolamine injection (Figure 3B). The memory-improving effect of EAE at 150 mg/kg bw was similar to that of donepezil (5 mg/kg bw), an acetylcholine esterase inhibitor. 

The Y-maze test showed that oral administration of EAE at a concentration of 150 mg/kg bw to mice improved the cognitive performance impaired by the scopolamine injection; however, this effect was marginal (Figure 3C). 

### 3.4. Effect of EAE on Lipid Peroxidation and DNA Damage in Mice Challenged with Scopolamine

Mice treated with scopolamine had a significantly higher serum MDA level than the untreated controls. Pretreatment of mice with the EAE (150 mg/kg bw) or AChE inhibitor donepezil (5 mg/kg bw) led to a significant reduction in the hepatic lipid peroxidation induced by scopolamine. In addition, the serum levels of 8-OHdG, a biomarker for oxidative stress, were also decreased in the group injected with scopolamine followed by feeding with EAE (100 or 150 mg/kg bw), compared to the experimental group challenged with the scopolamine alone (Figure 4).

### 3.5. Effect of EAE on the Expression of BDNF, GDNF, and NMDA Receptor in the Mouse Hippocampus

The transcription of the *BDNF* gene in the mouse hippocampus was significantly suppressed in the mouse group injected with scopolamine and was restored to normal levels by pre-exposure to EAE (100 or 150 mg/kg bw) or the AChE inhibitor donepezil at 1 h before scopolamine injection (Figure 5A). The mRNA level of the *GDNF* gene in the mouse hippocampal tissue that was slightly inhibited by the scopolamine injection seemed to recover to the untreated control levels by pre-exposure to EAE (100 or 150 mg/kg bw) or donepezil (Figure 5B). The expression pattern of the *NMDA receptor* transcript was similar to that of the *BDNF* and *GDNF* genes (Figure 5C).

### 3.6. Effect of EAE on the Nrf2/HO-1 Signaling Pathway and the PSD-95 Levels in the Hippocampus

Nuclear levels of Nrf2, an archetypical regulator of the antioxidant defense enzymes, in the hippocampal tissues from the mice pre-exposed to EAE at 150 mg/kg bw increased, compared with the control (Figure 6A). Furthermore, HO-1, a downstream gene product of Nrf2, in the mouse hippocampus, was also upregulated by treatment with EAE as well as donepezil (Figure 6B).

Meanwhile, the protein expression of *PSD-95*, which is reported to play a crucial role in synaptic plasticity, was significantly enhanced by the pretreatment with EAE at 150 mg/kg bw while it was not influenced by donepezil (5 mg/kg bw) or EAE at doses of 50 or 100 mg/kg bw.

### 3.7. Effects of the Oral Administration of EAE on the Scopolamine-Induced Neuronal Damage in the Hippocampus

The *cornu ammonis* (CA) regions consisting of CA1-4 in the hippocampus are considered to primarily dictate memory and cognition. In particular, CA1 pyramidal neurons are vulnerable to endogenous and exogenous oxidative stresses and are closely associated with the initiation of cognitive impairment. As depicted in Figure 7, the histological study showed that scopolamine injection damaged the CA1 region of the hippocampus as indicated by a partial disruption in the region. However, the pretreatment with EAE or donepezil abated the scopolamine-induced injury in the CA1 region.

## 4. Discussion

*Euonymus alatus* is a well-known plant utilized for medicinal purposes in some Asian countries for the treatment of illnesses such as cancer, stomach ache, wounds, asthma, hyperglycemia, and diabetic complications [20,21]. A preliminary study conducted by the authors showed that the extract of the *Euonymus alatus* leaf produced a moderate radical scavenging capacity as well as antioxidant enzyme-inducing activity (Figure 1 and Figure 8). As many of the antioxidant enzyme inducers were associated with the protective effects of neurons from oxidative stress, we hypothesized that the ethanolic extract of *Euonymus alatus* leaf had a neuroprotective effect and potentially exerted preventive action against the memory impairment induced by the ROS-generating chemicals such as glutamate and scopolamine [5,22]. As expected, the challenge of the mouse hippocampal HT22 cells with glutamate, a ROS generator in neuronal cells inhibited the glutathione synthesis, led to a significant reduction in cell viability, and this effect was effectively counterbalanced by EAE in a dose-dependent manner, suggesting neuroprotective potential for EAE via antioxidant activity. Furthermore, EAE also demonstrated in vitro antioxidant activity and antioxidant enzyme-inducing activity during the in vivo tests (Figure 6).

The EAE on the scopolamine-induced cognitive deficits was found to be effective in protecting mice from memory impairment induced by scopolamine (Figure 3). Therefore, to investigate the possible mechanism of action of EAE on cognitive improvement, the oxidative products were analyzed in the serum and the liver tissue. The levels of MDA and 8-OHdG, which were biomarkers for lipid peroxidation and DNA damage by oxidative stress, respectively, were significantly increased in the mice following the i.p. injection of scopolamine. However, these levels recovered to that of the untreated control by treatment with EAE, which further supported our hypothesis that EAE achieved cognitive improvement in mice through antioxidant activity. Regarding the use of scopolamine as an oxidative stress-inducing agent, several recent studies have reported that memory dysfunction in the scopolamine-treated animal model was related to enhanced oxidative stress within the brain [23,24,25]. Scopolamine, a well-known antagonist against the muscarinic cholinergic receptor, has been widely employed to study cognitive deficits in experimental animals. The i.p. injection of scopolamine is known to block cholinergic neurotransmission and subsequently lead to cholinergic malfunction and cognitive impairment in rats [25].

In addition, EAE induced the expression of BDNF, GDNF, and NMDA receptors in the mouse hippocampus (Figure 5). BDNF has recently been reported to activate the Nrf2 signaling pathway in the astrocyte [26]. It is most likely that EAE stimulated the Nrf2 signaling pathway and downstream antioxidant enzymes through the induction of BDNF. However, the possibility that EAE has improved the cognitive function of mice by regulating the BDNF independently of the Nrf2 signaling pathway cannot be excluded as the neurotrophin BDNF is associated with regulating the neuronal structure and function in both immature and mature central nerve systems. Furthermore, BDNF, which plays a key role in synaptic efficacy, has been shown to be involved in neuronal survival, synaptic plasticity, and memory [27] and reportedly promotes the survival of the basal forebrain cholinergic neurons [28,29,30]. Abnormal degeneration in both basal forebrain and cortex were generally thought to be important neuroanatomical markers in the AD patients [31,32]. Furthermore, a slower rate of cognitive dysfunction was closely related to higher BDNF serum levels in AD patients [33]. Thus, further study using careful experimental design using BDNF or the Nrf2 knockout mouse model was required to reveal the exact mechanism of action of EAE. The authors speculated that the antioxidant enzyme-inducing activity of EAE played a major role in neuroprotection and protection from scopolamine-induced memory impairment.

Meanwhile, the inhibitory activities of EAE, rutin, and quercetin against acetylcholine esterase were insignificant compared to donepezil, a known AChE inhibitor (Figure 9), eliminating the possibility that the neuroprotective and cognition-enhancing effects of EAE are mediated by AChE inhibition.

An issue to be addressed in future studies will be to identify the component(s) in EAE responsible for neuroprotection and cognitive improvement under the scopolamine insult. This study did not confirm the compound(s) that contributed to efficacy against cognitive decline. Instead, we measured the concentration of rutin and quercetin (flavonoids reported to be abundantly present in EAE) and found that EAE contained 1.07% rutin with a negligible level of quercetin. Interestingly quercetin and rutin did show a dose-dependent induction of the ARE-mediated transcriptional activation in the HT22-ARE cells. Furthermore, the activity of rutin was relatively mild (Figure 8), suggesting that rutin (quercetin-3-*O*-rutinoside) in EAE should have been converted to quercetin in the microflora in the large intestine and the resulting free quercetin promoted the Nrf2 signaling pathway and thereby protected the mice from scopolamine-induced memory impairment. It has been reported that rutin is absorbed slower than quercetin because it should be hydrolyzed into quercetin by cecal microflora, whereas quercetin is directly absorbed in the small and/or large intestine without further hydrolysis. Furthermore, quercetin is relatively stable in the body and can be easily maintained at a high concentration in plasma via a regular supply of rutin in the diet [34]. However, there is also a possibility that EAE contains other Nrf2 inducer(s) than rutin and quercetin. For instance, *Euonymus alatus* has been reported to contain sesquiterpenes, and electrophiles which might potentially induce the Nrf2 signaling pathway [35], and several other candidates including 9-epi-blumenol B, corchoionol C, and loliolide [20]. Thus, further study is required in order to identify the active compounds within this plant extract (EAE) that are involved in the improvement of the scopolamine-mediated cognitive decline.

## 5. Conclusions

In conclusion, ethanolic extract from the *Euonymus alatus* leaf was found to attenuate scopolamine-induced cognitive impairment in a mouse model potentially through the activation of the Nrf2 signaling pathway and via downstream antioxidant enzymes. Although rutin was abundantly present in this extract and was presumed to be responsible for the antiamnesic activity, further study to identify the active component(s) is required in the future.

## Figures and Tables

**Figure 1 antioxidants-09-00449-f001:**
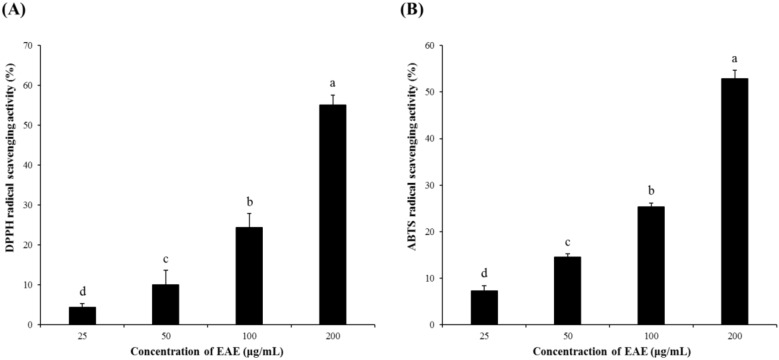
DPPH and the ABTS radical scavenging capacities of *Euonymus alatus* leaf (EAE). (**A**) DPPH and (**B**) ABTS radical scavenging activities. Results are expressed as mean ± SD (*n* = 3). Statistical analysis was performed using a one-way ANOVA, followed by Duncan’s multiple range test. Values with different alphabetical letters (a–d) represent significant differences from each other (*p* < 0.05).

**Figure 2 antioxidants-09-00449-f002:**
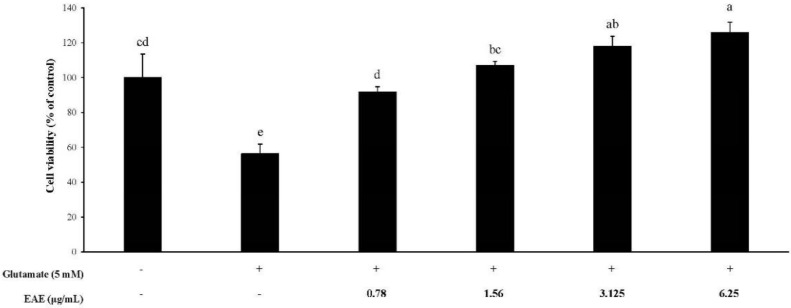
Protective effect of the EAE on glutamate-induced cell growth inhibition in HT22 cells. Results are expressed as mean ± SD (*n* = 3). Statistical analysis was performed using a one-way ANOVA, followed by Duncan’s multiple range test. Values not sharing common alphabetical letters (a–d) represent significant differences from each other (*p* < 0.05).

**Figure 3 antioxidants-09-00449-f003:**
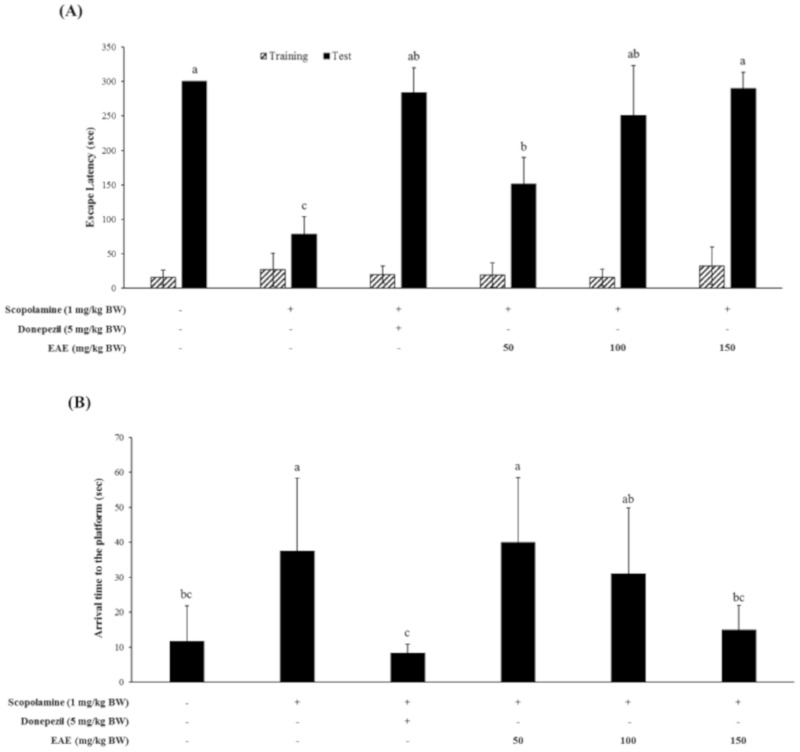
Effect of EAE on the scopolamine-induced memory deficit tested by the passive avoidance task (**A**), Morris water maze test (**B**), and Y-maze test (**C**). Results are expressed as mean ± SD (*n* = 5). Statistical analysis was performed using one-way ANOVA, following Duncan’s multiple range test. Bars not sharing common alphabetical letters (a–c) represent significant differences from each other (*p* < 0.05).

**Figure 4 antioxidants-09-00449-f004:**
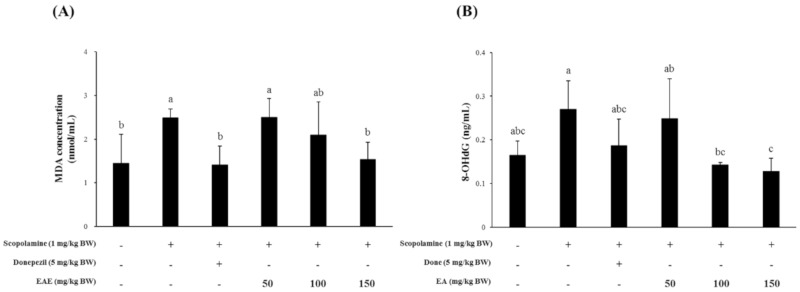
Supplementation with EAE decreased the oxidative damage on lipid and DNA in scopolamine-treated mice. (**A**) The MDA levels in the liver homogenate and (**B**) 8-OHdG levels in the serum. Results are expressed as mean ± SD (*n* = 3). Statistical analysis was performed using one-way ANOVA, following Duncan’s multiple range test. Bars not sharing common alphabetical letters (a–c) represent significant differences from each other (*p* < 0.05).

**Figure 5 antioxidants-09-00449-f005:**
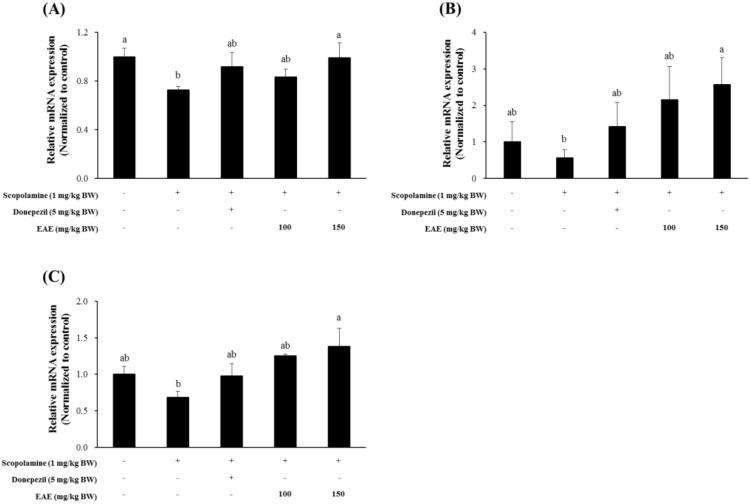
Effects of EAE on the mRNA expression of the *BDNF* (**A**), *GDNF* (**B**), and *NMDA receptor* (**C**) genes in the mouse hippocampus. Results are expressed as mean ± SD (*n* = 3). Statistical analysis was performed using one-way ANOVA, following Duncan’s multiple range test. Bars not sharing common alphabetical letters (a,b) represent significant differences from each other (*p* < 0.05).

**Figure 6 antioxidants-09-00449-f006:**
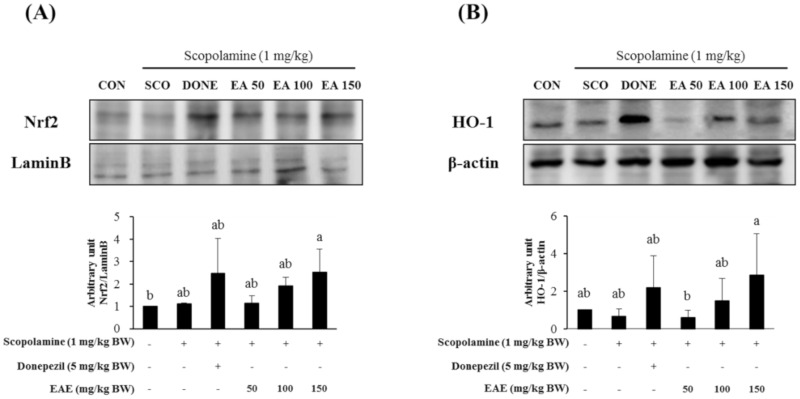
Effects of EAE on the protein levels of nuclear Nrf2 (**A**) and protein expression of HO-1 (**B**) and PSD-95 (**C**) in the mouse hippocampus. Results are expressed as mean ± SD (*n* = 3). Statistical analysis was performed using one-way ANOVA, following Duncan’s multiple range test. Bars not sharing common alphabetical letters (a,b) represent significant differences from each other (*p* < 0.05).

**Figure 7 antioxidants-09-00449-f007:**
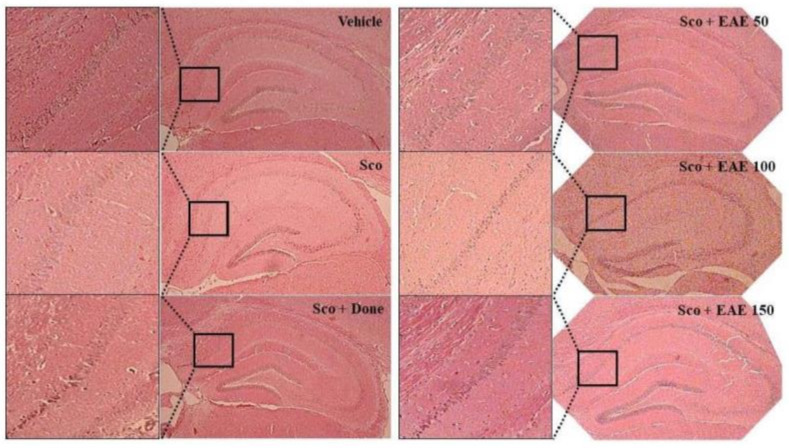
Prevention of scopolamine-induced neuronal damage in the hippocampal CA1 region by oral supplementation of EAE. The Vehicle, no treatment; Sco, *i.p* injection with scopolamine (5 mg/kg bw) alone, Sco + Don, treatment with donepezil followed by i.p. injection with scopolamine; Sco + EAE 50, treatment with EAE (50 mg/kg bw) followed by i.p. injection with scopolamine, Sco + EAE 100, treatment with EAE (100 mg/kg bw) followed by i.p. injection with scopolamine, Sco + EAE 150, treatment with EAE (150 mg/kg bw) followed by i.p. injection with scopolamine. A representative picture for each experimental group is depicted (magnification: 100× or 400×).

**Figure 8 antioxidants-09-00449-f008:**
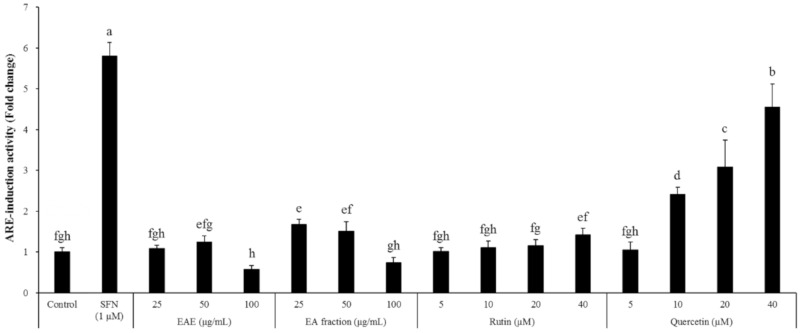
Effects of the EAE and EA fractions such as rutin and quercetin on ARE-mediated transcriptional activation in the HT22-ARE cells. Results are expressed as mean ± SD (*n* = 3). Statistical analysis was performed using one-way ANOVA, followed by Duncan’s multiple range test. Values not sharing common alphabetical letters (a–h) represent significant differences from each other (*p* < 0.05).

**Figure 9 antioxidants-09-00449-f009:**
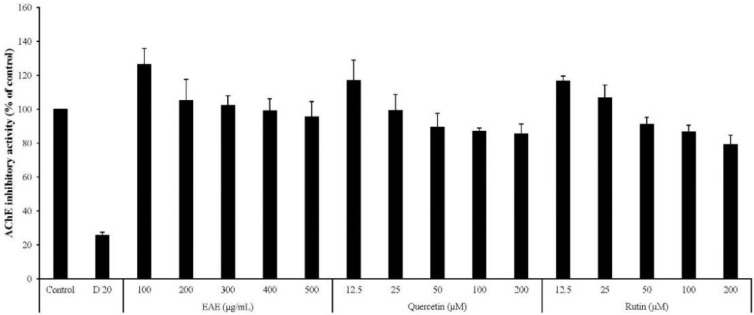
Acetylcholine-esterase-inhibitory activities of EAE.

**Table 1 antioxidants-09-00449-t001:** Experimental groups for animal behavioral study.

Experimental Group *	Scopolamine (1 mg/kg BW)	Treatment
Control	−	Vehicle
Scopolamine	+	Vehicle
Donepezil	+	5 mg/kg BW
EAE_Low	+	50 mg/kg BW
EAE_Middle	+	100 mg/kg BW
EAE_High	+	150 mg/kg BW

* *n* = 8 mice per group.

**Table 2 antioxidants-09-00449-t002:** Primer sets for real-time PCR.

Gene	Primer (5′→3′)		Annealing Temperature (°C)
	**Forward**	**Reverse**	
*BDNF* (NM_001316310)	CACTGGCTGACACTTTTGAGCAC	GCTGTGACCCACTCGCTAATACTG	62 °C
*GDNF* (NM_001301357)	CCCGCTGAAGACCACTCCCTC	GCGCTGCCGCTTGTTTATCTGG	64 °C
*NMDA Receptor1* (NM_001372559)	GCAGFAAACCAGGCCAATA	TGACAGGGCCATCTGTAT	54 °C
*β**-actin* (XM_030254057)	ACTATTGGCAACGAGCGGTT	ATGGATGCCACAGGATTCCA	52 °C

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
