# Peer review of "Neuroprotective Effects of Euonymus alatus Extract on Scopolamine-Induced Memory Deficits in Mice"

_antioxidants, 2020, doi:10.3390/antiox9050449_

Round 1

Reviewer 1 Report

This is an interesting work on putative neuroprotective effects of ethanolic Euonymus alatus extract on scopolamine-induced memory deficits in mice.
However, despite worthwhile elements there are certain issues that do not allow me to suggest this manuscript for the possible publication in the Journal.

1. First of all it is worth mentioning that more than 128 chemical constituents have been isolated and identified from E.alatus (see the Ref. 6; See also the Refs. 13 and 27). Moreover, the neuroprotective (basically the antioxidant) effects of EAE and certain constituents have been already analysed as well (e.g., doi: 10.1080/13880200902967987, doi: 10.1055/s-0036-1596352, or doi: 10.1039/C4FO00068D).
Thus, why do the authors decided to use the EAE instead of rutin and quercetin or even some other chemicals for the analysis of possible neuroprotective effects in mice model? What is the novelty of this work?

2. If the leaves of E.alatus are registered as edible in the Korean Food Code, therefore I wonder
how much (if the amount of consumed green material is known) of rutin or quercetin is consumed on regular basis? How well those numbers are represented by the chosen doses (50-150 mg EAE/kg BW) for the mice? The bioavailability of rutin and quercetin should be discussed as well (e.g., doi: 10.1691/ph.2012.2050; doi: 10.1016/s0014-5793(97)00467-5, doi: 10.1111/1541-4337.12342).

3. In addition, I suggest to provide data on the effects of EAE on normal mice (an additional control group treated with 150 mg/kg BW; it seems that only 150 mg/kg of EAE was the most effective: see the Figures 4A, 5 and 6).

4. What are the effects of EAE on viability of HT22 cells without glutamate?
What was a rationale to use such a range of EAE concentrations: 0.78-6.25 ug/mL? How well these
concentrations reflect the concentrations of EAE used for the experiments on mice (50-150 mg/mL BW)?
Since the main EAE components are rutin and quercetin (and kaempferol, as well), therefore it would be reasonable to know how much of those chemicals were present in 0.78-6.28 ug/mL of EAE. Can we expect the same results if rutin and/or quercetin could have been used instead of EAE?

5. Lines 289-290 (and the Fig. 5B): "The mRNA level of the GDNF gene in the mouse hippocampal tissue was also found to be inhibited by the scopolamine injection;" - I think this is not correct, since the statistical analysis of data did not confirm that. It is also questionable whether donepezil (line 291; Fig 5B) restored the expression of GDNF gene to the control level, as there are no statistically significant differences among control, donepezil and scopolamine treatments. Please, revise the interpretation of results presented in the Fig 5C as well (lines 291-292).

6. Lines 299-312: I think the interpretation of results slightly differs from the results we find in the
Fig 64-C.

Author Response

  1. First of all it is worth mentioning that more than 128 chemical constituents have been isolated and identified from E.alatus (see the Ref. 6; See also the Refs. 13 and 27). Moreover, the neuroprotective (basically the antioxidant) effects of EAE and certain constituents have been already analysed as well (e.g., doi: 10.1080/13880200902967987, doi: 10.1055/s-0036-1596352, or doi: 10.1039/C4FO00068D).

▶ Response: We are thankful for the reviewer’s valuable comments.

  • Ref 6, doi: doi:10.1155/2016/9425714, the review article entitled “Euonymus alatus: A Review on Its Phytochemistry and Antidiabetic Activity” refers to the hypoglycemic activity of alatus, rather than its neuroprotective effects. As the referee has pointed out, this review addressed “more than 128 chemical constituents have been isolated and identified from E. alatus”; but it did not specify the in vivo bioactive potential of those compounds. By contrast, our study demonstrated the neuroprotective and cognition-improving potential of the extract from the edible young leaf of E. alatus.
  • References 13 and 27: the articles just enumerate some compounds that were identified from alatus without presenting their relevant bioactive functions. In contrast, we focused on the cognition-improving potential of E. alatus leaf extract in this report.
  • References 10.1080/13880200902967987 addressed the in vitro neuroprotective effects of the alatus stem against Aβ insult, which is in line with our results. The difference between this paper and our study is that we used young leaves from E alatus (EA) which is critically important in terms of the safety issues associated with developing functional foods. According to the Korean Food Code, only young leaves from EA are permitted as an edible food ingredient. The novelty of our study is a clear demonstration of the in vivo efficacy of EA extract against scopolamine-induced cognitive impairment.
  • 1055/s-0036-1596352 reported several chemical compounds identified from EA and tested the in vitro neuroprotective effects of some of them. Among those compounds, botulin and methyl hydrogen tetradecanedioate showed antioxidant activity and exhibited cytoprotective effects against glutamate-induced death in HT22 cells.

à These results suggested that the neuroprotective activities of botulin and methyl hydrogen tetradecanedioate, but not rutin and quercetin, are associated with antioxidant activity. 10.1039/C4FO00068D reported the in vitro neuroprotective effect of kaempferol, which was isolated from the Petasites japonicus stem but not from any part of EA.

  1. Thus, why do the authors decided to use the EAE instead of rutin and quercetin or even some other chemicals for the analysis of possible neuroprotective effects in mice model? What is the novelty of this work?

▶ Response: We are thankful for the reviewer’s valuable comments.

The novelty of this work is the finding that EAE had an in vivo neuroprotective effect due to its antioxidative capability that allowed it to improve cognitive function. The results from our study are consistent with the previous reports, including those from other groups mentioned above. We believe our results further demonstrate the ability of EA to improve cognition in the mouse model of scopolamine-induced memory impairment.

From a pharmaceutical point of view, it would be worthwhile to elucidate the biological and/or physiological efficacy of purified compounds present in EAE, as the referee commented. However, from the practical point of view of the food industry, we believe it is also worth investigating the in vivo effects of EA extract because there should be synergistic interactions among its constituents. EA is presumed to contain numerous biologically active phytochemicals, including rutin and quercetin. In particular, only the leaf part of EA is registered as an edible ingredient in the Korean Food Code.

  1. If the leaves of E. alatus are registered as edible in the Korean Food Code, therefore I wonder how much (if the amount of consumed green material is known) of rutin or quercetin is consumed on regular basis? How well those numbers are represented by the chosen doses (50-150 mg EAE/kg BW) for the mice?

▶ Response: We are thankful for the reviewer’s valuable comments.

Since the mouse-to-human conversion factor is 12.3 (doi: 10.4103/0976-0105.177703), 50‒150 mg EAE/kg BW can be converted to 4−12 mg/kg BW, which is equivalent to 240−720 mg EAE per day in a person weighing 60 kg. However, a well-designed human study is required to validate this calculation and develop EA as a functional food. The final goal of this study is to provide information to justify a human study that, if successful, will allow us to develop EA as a functional food(s).

According to a study (doi: 10.3746/jkfn.2003.32.5.764), leafy vegetables are consumed quite frequently in Korea. They are consumed as much as 14.9 ± 7.8 g/meal on average, which accounts for about 20.6% of each dish group, even though there is seasonal variation. Also, since EA contains about 77% moisture (doi:10.3746/jkfn.2017.46.5.592), 1 g of EA is composed of about 230 mg dry matter, which is equivalent to 145 mg of EAE considering the extraction yield. Therefore, the dietary intake of 2 g EA can provide about 290 mg EAE equivalent theoretically, which is feasible for human consumption.

  1. The bioavailability of rutin and quercetin should be discussed as well (e.g., doi: 10.1691/ph.2012.2050; doi: 10.1016/s0014-5793(97)00467-5, doi: 10.1111/1541-4337.12342).

▶ Response: According to the literature, the oral administration of rutin is converted to quercetin in the serum while the bioavailability of rutin is slightly higher than quercetin (Manach C et al. Bioavailability of rutin and quercetin in rats. FEBS Lett. 409(1):12-6 (1997)

  1. In addition, I suggest to provide data on the effects of EAE on normal mice (an additional control group treated with 150 mg/kg BW; it seems that only 150 mg/kg of EAE was the most effective: see the Figures 4A, 5 and 6).

▶ Response: Our unpublished data implied that the doses of EAE at ≤ 500 mg/kg BW did not cause any toxicity in mice. Unfortunately, we do not have the data from behavioral tests using the mice treated with the condition as the referee mentioned – supplemented with only EAE at 150 mg/kg BW with no scopolamine. However, based on the dose-dependent effect of EAE shown in Figure 4−6, we can speculate that EAE alone would not cause adverse behavioral alterations.

  1. What are the effects of EAE on viability of HT22 cells without glutamate?

▶ Response: Without glutamate, EAE treatment slightly increased the viability of HT22 cells: Specifically, EAE (0.78–6.25 µg/mL) increased viability by 129%−145%.

  1. What was a rationale to use such a range of EAE concentrations: 0.78-6.25 ug/mL?

▶ Response: EAE at ≤ 100 µg/mL was not toxic to HT22 cells. The cell viability in the range of 6.25–50.0 µg/mL was in a plateau phase, suggesting that 6.25 µg/mL EAE or its constituents was presumably sufficient to overcome the glutamate toxicity in our experimental setting. However, further studies are needed for the identification and quantification of the major components responsible for the cytoprotective effects.

  1. How well these concentrations reflect the concentrations of EAE used for the experiments on mice (50-150 mg/mL BW)?

▶ Response: Arithmetically, it is estimated that 0.067 µg/mL of rutin is present in 6.25 µg/mL of EAE and 1.5 mg rutin in 150 mg EAE since EAE contains 1.07% rutin. This is mentioned in the Discussion section. It is estimated that oral administration of 100 mg EAE (or ~1 mg rutin) per kg BW will produce 20 μg quercetin per L plasma based on the previous bioavailability study (the intake of 530 mg rutin/kg BW yielded a plasma concentration of 10.57 mg/L quercetin), which is compatible with 2 µg EAE (or 0.02 μg quercetin) per mL culture medium) (Manach C et al. Bioavailability of rutin and quercetin in rats. FEBS Lett. 409(1):12-6 (1997)

  1. Since the main EAE components are rutin and quercetin (and kaempferol, as well), therefore it would be reasonable to know how much of those chemicals were present in 0.78-6.28 ug/mL of EAE. Can we expect the same results if rutin and/or quercetin could have been used instead of EAE?

▶ Response: According to the literature (doi: 10.4103/0253-7613.201016; doi: 10.1016/j.bbr.2010.09.027), rutin and/or quercetin are expected to exhibit comparable effects. However, as mentioned above, EAE contains a variety of bioactive compounds that may work in a synergistic or multifaceted manner in vivo. Thus, the health beneficial effects of EAE other than its neuroprotective and cognition-improving effects would be of interest for future studies.

  1. Lines 289-290 (and the Fig. 5B): "The mRNA level of the GDNF gene in the mouse hippocampal tissue was also found to be inhibited by the scopolamine injection;" - I think this is not correct, since the statistical analysis of data did not confirm that. It is also questionable whether donepezil (line 291; Fig 5B) restored the expression of GDNF gene to the control level, as there are no statistically significant differences among control, donepezil and scopolamine treatments. Please, revise the interpretation of results presented in the Fig 5C as well (lines 291-292).

▶ Response:

The sentences in Lines 289−291 were revised according to the statistical results.

  1. Lines 299-312: I think the interpretation of results slightly differs from the results we find in the Fig 6A-C.

▶ Response:

The sentences depicting Figure 6 were revised.

Reviewer 2 Report

This is an interesting paper describing the neuroprotective effects of E. alatus extracts on induced memory deficit in mice.

The treated argument is of importance to the field, the used experimental approach and the technical quality are of good quality. The manuscript is well written, and the data are correctly presented and discussed by the Authors.

In my opinion the manuscript can be accepted for publication without any further revisions.

Author Response

Thank you for such a positive review of our manuscript. We’ve tried to accordingly incorporate other referee’s comments into this manuscript to further improve it.

Reviewer 3 Report

The manuscript by Woo et al. “Neuroprotective Effects of Euonymus alatus Extract

3 on Scopolamine-Induced Memory Deficits in Mice” is an interesting manuscript and its content is related to the scope of Antioxidants; it could be accepted for publication after addressing the following comments

The authors should complete the description of DPPH and ABTS radical scavenging assays by adding the equation according to which the radical scavenging activity was calculated.  (see for example doi: 10.3390/antiox9030208)

In addition, the description of ABTS radical scavenging assay presents a mistake. In line 78 the authors wrote that the ABTS+• radical cation was mixed with potassium persulfate. This is wrong because in the assay the ABTS reagent is used and than converted to its radical cation ABTS+• by addition of sodium persulfate.

Line 84. The method used for the quantitative determination of Rutin and Quercetin was not adequately described.Did the authors use an internal standard?

Line 239. Can the authors give an explanation about the cell proliferation stimulated by EAE?

Some inaccuracies are present, for exemple in line 353:

Regarding the validity of scopolamine as an oxidative stress” it will be better to say “an oxidative stress-inducing agent”

Author Response

The manuscript by Woo et al. “Neuroprotective Effects of Euonymus alatus Extract on Scopolamine-Induced Memory Deficits in Mice” is an interesting manuscript and its content is related to the scope of Antioxidants; it could be accepted for publication after addressing the following comments

  1. The authors should complete the description of DPPH and ABTS radical scavenging assays by adding the equation according to which the radical scavenging activity was calculated. (see for example doi: 10.3390/antiox9030208)

▶ Response :

Lines 84−87: A new paragraph was inserted to clarify how to calculate the DPPH and ABTS radical scavenging activity.

  1. In addition, the description of ABTS radical scavenging assay presents a mistake. In line 78 the authors wrote that the ABTS+•radical cation was mixed with potassium persulfate. This is wrong because in the assay the ABTS reagent is used and than converted to its radical cation ABTS+•by addition of sodium persulfate.

▶ Response:

The sentences in Lines 77−83 describing the protocol for ABTS radical scavenging assay was revised to clarify the method.

  1. Line 84. The method used for the quantitative determination of Rutin and Quercetin was not adequately described. Did the authors use an internal standard?

▶ Response:

We have not used any internal standard. However, the standard curves for rutin and quercetin were obtained and we calculated the quantities of those constituents in the EAE by reference to the curve. To clearly describe the methods, the paragraph of subsection 2.4. was revised.

  1. Line 239. Can the authors give an explanation about the cell proliferation stimulated by EAE?

▶ Response :

The phrase “that the EAE had stimulated the neural cell proliferation.” was revised to “that the EAE had protected the neural cells from glutamate-induced cytotoxicity.” since there is no solid evidence showing the proliferative effects of EAE in HT22 cells.

  1. Some inaccuracies are present, for exemple in line 353: “Regarding the validity of scopolamine as an oxidative stress” it will be better to say “an oxidative stress-inducing agent”

▶ Response:

The phrase was revised accordingly. Also, the expressions and descriptions throughout the manuscript were double-checked and revised to improve the readability and accuracy. The revised manuscript was proofread again.

Round 2

Reviewer 1 Report

Thank you for the explanations and answers you provided.